# Effects of Probiotics on Autism Spectrum Disorder in Children: A Systematic Review and Meta-Analysis of Clinical Trials

**DOI:** 10.3390/nu15061415

**Published:** 2023-03-15

**Authors:** Xiao He, Wenxi Liu, Fengrao Tang, Xin Chen, Guirong Song

**Affiliations:** 1Department of Health Statistics, School of Public Health, Dalian Medical University, No. 9 South Road, Lvshun District, Dalian 116044, China; he__xiao@163.com (X.H.); liuwenxi990903@163.com (W.L.); fengraot@163.com (F.T.); 2Department of Epidemiology, School of Public Health, Dalian Medical University, No. 9 South Road, Lvshun District, Dalian 116044, China; chenx@dmu.edu.cn

**Keywords:** probiotics, children, autism spectrum disorder, behavioral symptoms, meta-analysis

## Abstract

Many studies have explored the efficacy of probiotics on autism spectrum disorder (ASD) in children, but there is no consensus on the curative effect. This systematic review and meta-analysis aimed to comprehensively investigate whether probiotics could improve behavioral symptoms in children with ASD. A systematic database search was conducted and a total of seven studies were included in the meta-analysis. We found a nonsignificant overall effect size of probiotics on behavioral symptoms in children with ASD (SMD = −0.24, 95% CI: −0.60 to 0.11, *p* = 0.18). However, a significant overall effect size was found in the subgroup of the probiotic blend (SMD = −0.42, 95% CI: −0.83 to −0.02, *p* = 0.04). Additionally, these studies provided limited evidence for the efficacy of probiotics due to their small sample sizes, a shorter intervention duration, different probiotics used, different scales used, and poor research quality. Thus, randomized, double-blind, and placebo-controlled studies following strict trial guidelines are needed to precisely demonstrate the therapeutic effects of probiotics on ASD in children.

## 1. Introduction

Autism spectrum disorder (ASD) is a developmental disability caused by differences in the brain and is characterized by a series of neurodevelopmental disorders, including language and social disorders, restricted interests, and repetitive stereotyped activities [1]. At present, there is still no consensus on the specific cause of ASD, and it may be caused by a combination of genetic and environmental factors [2,3,4,5,6]. Furthermore, ASD has directly and indirectly affected many aspects, such as health, education, housing, and employment, and consequently has brought a serious burden to families and society. It is predicted that by 2025, the direct medical and nonmedical productivity costs of ASD in the United States alone will reach approximately $500 billion [7]. The prevalence of ASD is gradually increasing, with the prevalence of ASD in 8-year-olds in the United States increasing from 6.7 per 1000 in 2000 to 23 per 1000 in 2018 [8]. In 2019, the age-standardized prevalence of ASD in Chinese men and women was 585.32 per 100,000 and 142.75 per 100,000 [9]. In addition, prevalence data from both the United States and China suggest that the prevalence of ASD is higher in men than in women [8,9].

Despite the serious economic and social burden of ASD, there is currently no standard treatment for a complete cure due to its complex aetiology and diverse symptoms [10]. The treatment of ASD can be roughly divided into behavioral interventions and drug treatments, and the most widely used behavioral intervention is applied behavior analysis (ABA). In the latest National Clearinghouse on Autism Evidence and Practice (NCEAP) report, a systematic review of studies using behavioral interventions between 1990 and 2017 suggests that most behavioral interventions have some impact on children with ASD in different age groups [11]. Although there have been many experiments exploring the use of drugs to treat children with ASD, and some experiments have shown some therapeutic effects, there is no complete and more reasonable evidence to support their efficacy [12,13,14]. The increasing incidence of ASD and the limited efficacy of behavioral interventions and pharmacotherapy in children with ASD make the search for new treatments an urgent need.

Given the relatively high prevalence of gastrointestinal (GI) disorders in patients with ASD and the theory of the gut–brain axis (GBA), many studies have explored the therapeutic effects of probiotics on ASD. The GBA is an important pathway for mammalian gastrointestinal and central nervous system information exchange [15]. The GBA consists sympathetic and parasympathetic branches of the central nervous system, neuroendocrine system, autonomic nervous system, enteric nervous system, gut and gut microbes, and their metabolites. Gut microbes transmit information through the enteric nervous system to the vagus nerve and then reach the central nervous system to affect psychological, cognitive, behavioral and neurological functions [16].

Probiotics are live microorganisms that, when administered in adequate amounts, confer a health benefit on the host [17]. Because they produce and transport neuroactive substances and act on the gut–brain axis, such as γ-aminobutyric acid (GABA) produced by *Lactobacillus brevis* and *Bifidobacterium denticola*, and dopamine produced by *Staphylococcus aureus* and *Escherichia coli* [18], Dinan et al. defined them as psychobiotics [19]. Many species of probiotics have been used to treat ASD, including some single-strain probiotics, probiotic blends of different formulas, and probiotic products, which are often used in combination with dietary interventions or behavioral interventions. *Lactobacillus reuteri* and *Lactobacillus plantarum* have been shown to significantly improve social behavior in mice [20,21], but there is insufficient evidence to determine that probiotics have positive therapeutic effects on ASD in children.

Moreover, by a systematic literature search, we found that there is currently a lack of meta-analyses on the efficacy of probiotic treatment for children with ASD, and only one meta-analysis was found, in which only three studies were included for a pooled analysis [22]. Therefore, it is necessary to expand the scope of the literature search to include more studies on probiotic treatment for children with ASD and conduct a systematic review and meta-analysis to provide new ideas for the probiotic treatment of ASD.

## 2. Materials and Methods

The systematic review and meta-analysis were both directed in accordance with the preferred reporting items for systematic reviews and meta-analyses (PRISMA) statement. The protocol of this study was registered in PROSPERO on 10 May 2022 (registration number: CRD42022327948).

### 2.1. Search Strategies

A systematic literature search was performed in CNKI, PubMed, Cochrane Library, Web of Science, ScienceDirect, and Medalink by 9 March 2023, using search items such as (“autism spectrum disorder” OR “ASD” OR “autism” OR “autistic disorder” OR “Asperger syndrome” OR “Asperger disorder” OR “autistic traits”) AND (“microbiota” OR “microbiome” OR “microflora” OR “probiotic” OR “probiotics”). The detailed search strategies are listed in Appendix A.

### 2.2. Inclusion Criteria and Exclusion Criteria

The inclusion criteria were established according to the principles of PICOS (participants, intervention, comparison, outcome, study).

The inclusion criteria were as follows: (1) children and adolescents under 18 years old diagnosed with ASD or autistic disorder or Asperger syndrome or Asperger disorder according to the generally accepted standard; (2) the use of probiotic, probiotics or probiotic preparations as the main intervention in the trial group; (3) trials with no restrictions on control measures; (4) scores for autism-related behavioral symptoms measured by an eligible questionnaire; and (5) randomized controlled trials (RCT) and crossover trials.

Exclusion criteria of systematic review: (1) participants older than 18 years and (2) full sets of the literature that were still not available even after we contacted their authors.

Exclusion criteria of meta-analysis: (1) participants older than 18 years; (2) full sets of the literature that were still not available even after we contacted their authors; (3) the data required for meta-analysis were not available; and (4) the information on probiotic strains was not available.

### 2.3. Study Selection and Data Extraction

Initially, two researchers (He X and Tang FR) independently screened the literature retrieved through an evaluation of the titles and abstracts to select the literature that met our criteria. Then, the two researchers continued to independently appraise these selected studies by reading the full-text articles and extracted data, including the basic characteristics of study participants, sample size, intervention measures and comparison measures, intervention duration, scores for autism-related behavioral symptoms, scores for GI symptoms, etc.

Any divergence between the two researchers during the process of screening and data extraction was resolved by discussion or consulting a third reviewer (Song GR).

### 2.4. Study Quality Assessment

According to the PRISMA statement, the risks of bias of RCTs and crossover trials were evaluated from the following seven aspects: (1) random sequence generation; (2) allocation concealment; (3) blinding of participants and personnel; (4) blinding of outcome assessment; (5) incomplete outcome data; (6) selective reporting; and (7) other bias. Each bias was divided into three levels: low risk, unclear risk, and high risk.

### 2.5. Data Analysis

Review Manager 5.3 was used to evaluate the risk of bias for the RCT and crossover trials. All meta-analyses and visualizations were conducted by Review Manager (5.3) STATA/SE software (15.1) and the package “Meta-Analysis”. The means and standard deviations (SDs) of the change in scores for ASD-related behavioral symptoms between baseline and ending point (in the subsequent article just referred to as a “change in score”) were extracted from each intervention and control group of the included studies. The SDs of the changes in scores for some studies were not available directly from the source or authors; they were estimated based on the SDs of the scores at baseline and the ending point along with an assumed correlation coefficient of 0.5, according to the formula recommended in the Cochran handbook. Then, the standardized mean difference (SMD) was calculated with 95% confidence intervals (CIs) by Hedges’ method to assess effect size. The *I*^2^ statistic and *p* value of Cochran’s Q test were used to evaluate heterogeneity among studies, with values of <25%, 25–50% and >50% representing low, moderate, and high degrees of heterogeneity, respectively, and *p* < 0.05 was considered statistically significant heterogeneity. The fixed-effects model was used if *I*^2^ < 50%; otherwise, the random-effects model was applied. The Begg test and Egger test were used to assess publication bias.

Subgroup analyses were conducted to explore the sources of heterogeneity. The country of study, scales used, intervention measures, duration of intervention, and types of study were considered as the potential subgroup basis. Sensitivity analyses were performed to confirm the robustness of the results by removing one study and repeating the meta-analysis. All tests were two-sided and *p* < 0.05 was considered statistically significant.

## 3. Results

### 3.1. Literature Search

The process of the literature search and selection are shown in Figure 1. Initially, a total of 676 studies were retrieved, and 37 studies were selected after screening titles and abstracts. Of the thirty-seven studies: the full-text was not available in six of them; we actually found no evidence of the use of probiotics or probiotic preparations as the main intervention in twenty of them; and only one study included participants over eighteen years of age. However, of the remaining ten studies included for systematic review, three were not included in the final analysis due to the lack of availability of the relevant data. As a result, only seven studies ended up being included in our meta-analysis.

### 3.2. Study Characteristics

The characteristics of the 10 eligible studies are summarized in Table 1. Overall, 460 children aged 2–16 years with ASD from five countries (four studies from China [23,24,25,26], one from Italy [27], three from America [28,29,30], one from Britain [31], and one from Egypt [32]) were included in the systematic review. Six studies were RCTs [23,24,25,26,27,32], and four studies were crossover controlled trials of a completely randomized design [28,29,30,31].

All studies used probiotics as their interventions, three of which used a single-strain probiotic for the intervention [23,29,31], and the seven other studies used multiple-strain probiotics [24,25,26,27,28,30,32]. Overall, 15 probiotics were involved in the intervention, i.e., *Bifidobacterium longum*, *Lactobacillus acidophilus*, *Enterococcus faecalis*, *Lactobacillus plantarum*, *Streptococcus thermophilus*, *Bifidobacterium brevis*, *Bifidobacterium infantis*, *Lactobacillus para-casei*, *Lactobacillus delbrueckii* subsp., *Bulgaricus*, *Lactobacillus paracasei LPC-37*, *Bifidobacterium lactis BL-04*, *Lactobacillus acid ophilus*, *Lactobacillus deuteri*, and *Lactobacillus fermentum*. With the exception of the study by Niu et al. [24], all interventions included *Lactobacillus* species, and *Lactobacillus plantarum* (*n* = 5) [23,27,28,30,31], *Lactobacillus infantis* (*n* = 5) [26,27,28,29,30], and *Bifidobacterium longum* (*n* = 4) [25,27,28,30] were the three most commonly used probiotics. Probiotic doses used in all studies were reported. The doses of probiotics used in the 10 studies ranged from 9 × 10^7^ colony forming units (CFU) [25] to 9 × 10^12^ CFU [30], with more than half of the studies (*n* = 5) [23,24,26,27,31] using tens of billions of probiotic doses. In addition, the study by Arnold et al. divided patients into two different dose groups based on the duration of intervention, with a daily probiotic dose of 4.5 × 10^12^ CFU in the first four weeks and 9 × 10^12^ CFU in the last four weeks [30].

Of the 10 studies, six studies only used probiotics or probiotic preparations as interventions [23,27,28,30,31,32], two studies applied ABA interventions simultaneously [24,25], one added prebiotics, such as fructo-oligosaccharide [26], and the remaining study added bovine colostrum product (BCP) [29]. For control groups, only six studies used placebo [23,26,27,28,30,31], only two studies used ABA [24,25], one study used BCP [29], and one study used standard treatment [32]. The duration of intervention in the studies ranged from four weeks to six months, with eight studies having an intervention duration of three months or less [23,24,25,28,29,30,31,32], and the remaining two studies all having an intervention duration of greater than three months [26,27].

Five studies used the autism treatment evaluation checklist (ATEC) to measure changes in ASD severity and to evaluate the effects of treatment. ATEC consists four subscales, i.e., speech/language/communication, sociability, sensory/cognitive awareness, and health/physical/behavior [33]. Two studies used the aberrant behavior checklist (ABC), which was used to assess the effects of drugs and other treatments on severely developmentally disabled individuals [34]. The ABC consists five subscales, i.e., irritability, hyperactivity/noncompliance, lethargy/social withdrawal, stereotypic behavior, and inappropriate speech [35]. The remaining three studies used the autism behavior checklist-Taiwan version (ABC-T), clinical global impression-improvement (CGI-I), total autism diagnostic observation schedule-calibrated severity score (ADOS-CSS), and the development behavior checklist (DBC). The ABC-T is a 47-item questionnaire used to assess behavioral problems in children with intellectual and developmental disabilities, which is divided into five subscales: sensory (sensation and perception), relating (relation and connection), body and object use (physical activity and rigid use of objects), language (communication and interaction), and social and self-help (adaptability and self-care) [23]. Clinical global impression (CGI) is rated by a clinician to assess clinical efficacy, which consists clinical global impression-severity (CGI-S), clinical global impression-improvement (CGI-I), and clinical global impression-efficacy index (CGI-EI) [23]. The ADOS-CSS is used to quantify autism symptoms and is a standardized calibrated severity score of the autism diagnostic observation schedule [36]. The DBC is completed by a guardian to assess behavioral/emotional disorders and consists five subscales: (1) disruptive/antisocial behavior, (2) self-absorbed behavior, (3) communication, (4) anxiety problems, and (5) social problems [31]. Nine studies measured GI symptoms or GI microbial abundance [24,25,26,27,28,29,30,31,32]. Three studies reported adverse reactions [24,27,30], and six studies reported reasons for the withdrawal of participants [23,26,27,28,29,31]. Most of the studies had small or moderate sample sizes from 8 to 100. Otherwise, all studies were published in the last decade, except the study by Parracho et al. [31].

Five studies reported that probiotics significantly improved ASD-related symptom scores [23,24,25,26,32], while five other studies did not find such a significant improvement [27,28,29,30,31]. The studies by Wang et al. and Mohsen et al. found that probiotics had a certain improvement effect on gastrointestinal symptoms in children with ASD [26,32].

Niu et al. and Wang et al. found that there were significant differences in intestinal flora between children with and without ASD [24,26]. Niu et al. found that the abundances of *Bacteroides* and *Actinomycetes* in the intestines of children with ASD were significantly lower than those of children without ASD, regardless of whether the children had GI symptoms. The abundance of *Proteobacteria* was significantly higher in children with ASD than in children without ASD [24]. Similarly, the results of fecal bacterial profiling by Wang et al. showed that the abundance of *Clostridium* and *Ruminococcus* in the feces of children with ASD increased, but the abundance of *Bifidobacteriales* and *Bifidobacterium longum* decreased compared to children without ASD [26]. In addition, Wang et al. found that the concentrations of acetic acid, propionic acid, and butyric acid significantly declined in the feces of children with ASD, which indicated a change in the intestinal flora. The results of Niu et al. and Wang et al. both showed that children with autism had a lower abundance of beneficial flora and an increased abundance of harmful flora than children without ASD, and these results were consistent with previous studies [37]. In addition, the studies by Parracho et al. and Li et al. suggested that probiotics had a regulatory effect on the gut flora of ASD [25,31].

### 3.3. Study Quality

RCTs and crossover controlled trials of completely randomized design (*n* = 10) were evaluated by the seven dimensions recommended in the PRISMA guidelines. All studies (*n* = 10) were at low risk of bias for reporting measurements and other biases; most studies (*n* = 9) were at low risk of bias for attrition bias. However, most studies performed poorly on measurement bias; only one study [30] had a low risk of bias, and two studies [27,31] had an even higher risk of bias due to unreported or unblinded measurement personnel. Taken together, only one study [30] was of high quality, while two studies [27,31] were of low quality, and the rest were of average quality. Overall, these reviewed studies were of average quality. Detailed results are shown in Figure 2 and Figure 3.

### 3.4. Results of the Meta-Analysis

#### 3.4.1. Effects of Probiotics on Autism-Related Behavioral Symptoms of Children with ASD

A total of seven studies were included in the meta-analysis and were all published in nearly five years. A total of 268 children aged 1.5 to 15 years with ASD were included in these studies from three countries, China (*n* = 3), Italy (*n* = 1), and America (*n* = 3). In addition, a total of 178 children with ASD were assigned to the intervention group and 172 children to the control group. The characteristics of the seven studies are shown in Table 2. There was moderate heterogeneity between studies (*p* < 0.05, *I*^2^ = 54%), and so the random-effects model was used. The overall difference in the improvement of autism-related behavioral symptoms between the intervention group and the control group was not statistically significant (pooled SMD = −0.24, 95% CI: −0.60 to 0.11, Z = 1.34, *p* = 0.18). The forest plot shows the results of the pooled analysis in Figure 4.

#### 3.4.2. Subgroup Analyses

None of the subgroup analyses according to different countries, scales, and durations of intervention showed significantly different improvements in ASD-related behavioral symptom scores between the intervention and control groups. Detailed information on these subgroup analyses is displayed in Appendix A.

Additionally, in the subgroup of single-strain probiotics, the overall difference in the improvement in ASD-related behavioral symptom scores between the intervention and control groups was not statistically significant (pooled SMD:0.30, 95% CI: −0.57 to 1.16, *p* = 0.50). In the subgroup of probiotic blends, the intervention group showed a significant improvement in ASD-related behavioral symptom scores compared to the control group (pooled SMD: −0.42, 95% CI: −0.83 to −0.02, *p* = 0.04), as shown in Figure 5.

Furthermore, the studies were divided into two groups, RCTs and crossover trials, according to the type of study. In the RCT subgroup, the overall difference in the improvement in ASD-related behavioral symptom scores between the intervention and control groups nearly approached statistical significance (pooled SMD: −0.45, 95% CI: −0.94 to 0.03, *p* = 0.07), whereas in the crossover trial subgroup, a significant difference was not found (pooled SMD: 0.04, 95% CI: −0.55 to 0.63, *p* = 0.89), as shown in Figure 6.

#### 3.4.3. Publication Bias and Sensitivity Analysis

As fewer than 10 studies (*n* = 7) were included in this meta-analysis, Begg and Egger tests were used to assess publication bias instead of funnel plots, suggesting a low likelihood of publication bias (*p* > 0.05). To assess the stability of the meta-analysis results, sensitivity analysis was performed on the seven included studies. The overall effect size did not change significantly after any one study was excluded; therefore, the results of this meta-analysis were relatively robust.

Detailed information on the tests of publication bias and sensitivity analysis are displayed in Appendix A.

## 4. Discussion

This review was an update to a previous meta-analysis that only included clinical controlled trials to explore whether probiotics could improve the overall severity of ASD symptoms in children [21]. We expanded the literature search, included more studies, and provided an overview of studies with a wider variety of probiotics or probiotic preparations in the treatment for children with ASD. However, only 10 eligible studies were included in the systematic review, which involved six RCTs [23,24,25,26,27,32] and four crossover controlled trials using a completely randomized design [28,29,30,31]. Most of these studies were of average quality and only one was of high quality [30]. Moreover, the interventions in these studies involved a total of 15 probiotics, such as *Lactobacillus plantarum*, *Lactobacillus infantis,* and *Bifidobacterium longum*. The intervention duration ranged from four weeks to six months and a variety of scales for measuring ASD-related symptoms and GI symptoms in these studies were very heterogeneous. Therefore, there is still a dearth of high-quality studies to test the therapeutic effects of probiotics on ASD symptoms in children.

Although animal studies showed that probiotic supplementation improved ASD-like symptoms and social behavior in mice [38,39], in our meta-analysis, the pooled SMD (−0.24, 95% CI: −0.60 to 0.11, *p* = 0.18) showed that probiotic supplementation did not improve the associated behavioral symptoms in children with ASD, which is consistent with previous findings [37,40]. The pooled SMD in each subgroup by country, scale, and duration of intervention also showed the same results. We conjecture that the results of animal models and human experiments may be different for the following reasons: (1) mice and humans are different species and there are large species differences; (2) ASD model mice are artificially induced by drugs and other means, the causes of ASD in humans are complex and diverse [41], and the specific mechanism has not yet been elucidated; and (3) some tests used to evaluate mouse-related indicators cannot be used on humans for ethical reasons [42]. Therefore, more research is needed in the future to explore the extrapolation of the use of probiotics for autism from animals to humans.

Interestingly, the pooled SMD (−0.42, 95% CI −0.83 to −0.02) in the subgroup of the probiotic blend intervention showed that the probiotic blend significantly improved the associated behavioral symptoms in children with ASD. Furthermore, the probiotic blend intervention showed a more significant improvement than the single-strain probiotic intervention (pooled SMD = 0.30, 95% CI −0.57 to 1.16) in the associated behavioral symptoms in children with ASD. This may be due to the simultaneous effects of different probiotics through different pathways and the possible interaction between their metabolites, which can enhance the original neuromodulatory effect. However, there is currently insufficient clinical evidence to support the idea that probiotic blends have better therapeutic outcomes [43]. In addition, we noted that the probiotic mixture included in this meta-analysis contains probiotics of the genus *Lactobacillus*, and the study by Buffington et al. found that *Lactobacillus reuteri* significantly improved the social ability of mice [44], so we speculate that *Lactobacillus reuteri* has a similar effect on humans. Some studies have found that GABA produced by *Lactobacillus* is the main inhibitory neurotransmitter in the central nervous system and has a certain regulatory effect on intestinal neurosecretory neuropeptides and the immune system [18]; thus, the therapeutic value of probiotics on ASD in children warrants research.

It is also worth mentioning that the pooled SMD (−0.45, 95% CI −0.94 to 0.03, *p* = 0.07) in the RCT subgroup nearly approached statistical significance, but a significant improvement in the associated behavioral symptoms in children with ASD did not appear in the subgroup of crossover trial. The pooled SMD of the RCTs suggests that probiotic supplementation might have a potential therapeutic effect on behavioral symptoms associated with ASD in children. Each crossover trial included in this meta-analysis had a much shorter washout period (the longest washout period was only four weeks) that it did not completely eliminate the effects of the intervention taken at the previous stage, and of conditioning effects in the second (placebo) phase, while the initial treatment was the probiotic [43]. Thus, more RCTs with high quality are needed to explore and validate the hypothesis that probiotics might have potential therapeutic effects on behavioral symptoms of ASD in children [45].

Of note, although neither of the two subgroups defined by the duration of intervention showed a significant improvement in the behavioral symptoms associated with ASD in children, the pooled SMD (−1.09, 95% CI −2.78 to 0.59, *p* = 0.20) from the subgroup with an intervention duration greater than three months was more likely to show improvement than that from the subgroup with an intervention duration of less than three months (pooled SMD: −0.11, 95% CI −0.45 to 0.23). This result suggests that the therapeutic effect of probiotic supplementation in children with ASD may be more pronounced with increasing intervention time. In the study by Wang et al. [26], the intervention group showed no significant improvement (*p* > 0.05) in ATEC total score and ATEC subscale score after 30 days of probiotic and fructo-oligosaccharide intervention, while after 60 days and 108 days of intervention, ATEC total score, speech/language/both communication scores, and sociability scores improved significantly (*p* < 0.05) from baseline, but no significant improvements (*p* > 0.05) were found in the control group at 30, 60, and 108 days after receiving a placebo. In addition, existing studies have found that measurable changes in behavior can occur after 2 weeks of probiotic supplementation in animals and 4 weeks after probiotic supplementation in humans [43], and so, in future studies, the intervention duration should be appropriately extended to achieve more obvious expected effects.

Interestingly, although no significant improvement was observed in the control and intervention groups aged 13–15 years, the Chinese version of the Swanson, Nolan, and Pelham-IV (SNAP-IV) scores improved significantly in all aspects in the 7–12 year group in the intervention group [23]. These results suggested that probiotics may be more effective in younger children with ASD, as is consistent with the results of the study by Oono et al. [46] and Reichow et al. [47] for the early intervention in children with ASD. The onset of ASD might occur in infancy and early childhood prior to 36 months of life, and the younger the child, the shorter the course of the disease and the milder the symptoms; thus, it is believed that early intervention and treatment for children with ASD could achieve better effects.

Previous studies have found that there were significant differences in intestinal flora between children with and without ASD [24,26,37] and that children with ASD have fewer beneficial bacteria and more harmful bacteria. Similarly, Iovene et al. and Ristori et al. found that the relative abundance of fecal microorganisms in children with ASD was also different from that of normal children, and *Candida spp.* in the feces of children with ASD was significantly more abundant than that of normal children (*p* = 8.67 × 10^−6^) [48], but the abundance of *Prevotella*, *Coprococcus*, *Enterococcus*, *Lactobacillus*, *Streptococcus*, *Lactococcus*, *Staphylococcus*, *Ruminococcus*, and *Bifidobacterium* species was lower [49]. In addition, another study has demonstrated that transplanting the gut microbiota of human patients with ASD into mice can induce signature ASD behaviors in mice [41]. Therefore, we speculate that changes in the gut and fecal microorganisms may be one of the pathogenic mechanisms of ASD. In a systematic review by Martínez-González et al., seven studies found a statistically significant difference in GI symptoms between children with ASD and children without ASD, and they also found that GI symptoms in children with ASD were strongly associated with the severity of neurobehavioral disorders [23]. Studies by Wang et al. and Mohsen et al. confirmed that probiotic supplementation was effective in improving GI symptoms in children with ASD [26,32]. Additionally, studies by Li et al. found [25] that the relative abundance of beneficial bacteria (*Bifidobacterium* and *Lactobacillus*) in the gut of children with ASD increased after probiotic supplementation. Children with ASD have obvious selective and over-particular eating behaviors, which can display a certain impact on GI symptoms and intestinal flora abundance [50]. In summary, it was speculated that probiotics could improve GI symptoms in children with ASD by increasing the relative abundance of beneficial bacteria in the gastrointestinal tract. The improvement of GI symptoms by *Bifidobacterium* and *Lactobacillus* may be related to butyric acid in their metabolites, which has a restorative effect to a certain degree on the barrier function of the intestinal epithelial mucosa [25].

### Strengths and Limitations

This study included more of the literature and performed more detailed subgroup analyses than previous studies; therefore, our study could more comprehensively aid in exploring the therapeutic effects of probiotics on children with ASD. Nevertheless, our study still had several potential limitations. First, only six databases were searched, the grey literature was not included, and the number of currently included studies may be smaller than the actual total number of eligible studies. Second, as only seven studies were included in this meta-analysis, it was impossible to effectively explore the sources of heterogeneity. Third, the studies included in the meta-analysis were only from three countries, and each study had a small sample size, which may consequently weaken the persuasiveness of our findings. Fourth, original data in several studies were not obtained, and so some measures were estimated based on the methodology recommended by the Cochrane handbook, which undoubtedly had an impact on the reliability of the results. Finally, most of the included studies did not report side effects, so we were unable to provide a valid assessment of the safety of probiotic treatment for children with ASD.

## 5. Conclusions

In conclusion, positive significant effects of probiotics were not observed in children with ASD in our meta-analysis. The studies used provided limited evidence for the efficacy of probiotics on children with ASD due to their small sample sizes, shorter intervention duration, different probiotics used, different scales used, and poor research quality. Significantly, multiple-strain probiotic blend intervention exhibited a positive therapeutic effect on children with ASD and was more effective than single-strain probiotics in subgroup analyses. Moreover, subgroup analyses suggested that studies with longer intervention durations and RCT designs might be more likely to reveal the effects of probiotic treatment in improving ASD-related behavioral symptoms in children. In short, to demonstrate the therapeutic effects of probiotics on children with ASD, randomized, double-blind, and placebo-controlled studies following strict trial guidelines are needed. At the same time, researchers should consider the different species in probiotics, the different ages of children with ASD, the different GI symptoms of children with ASD, and the different intervention durations to conduct related research and provide more precise and credible evidence.

## Figures and Tables

**Figure 1 nutrients-15-01415-f001:**
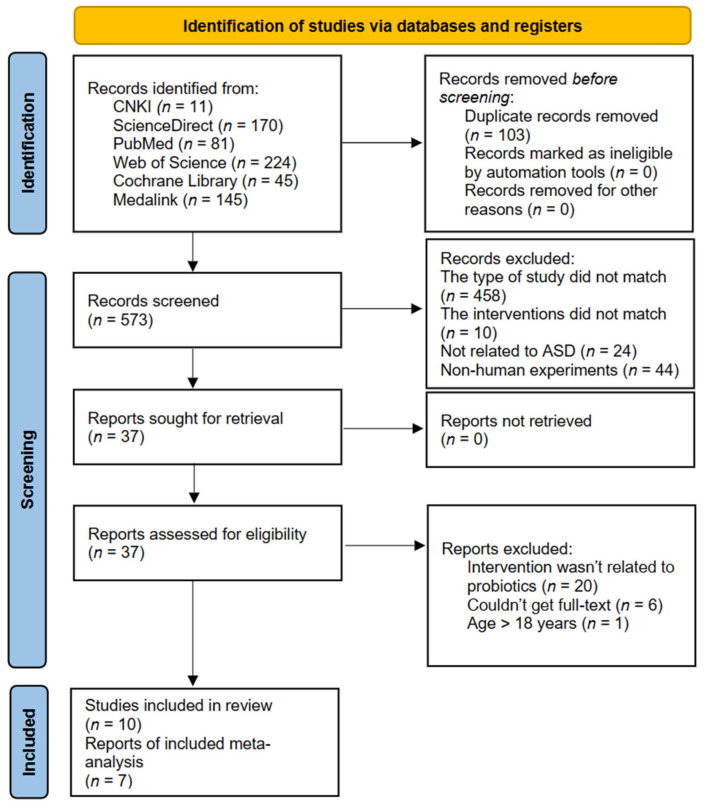
PRISMA flow diagram.

**Figure 2 nutrients-15-01415-f002:**
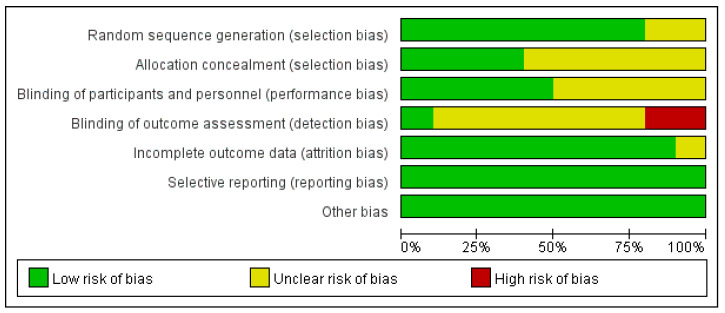
Risk of bias graph for RCTs and crossover controlled trials.

**Figure 3 nutrients-15-01415-f003:**
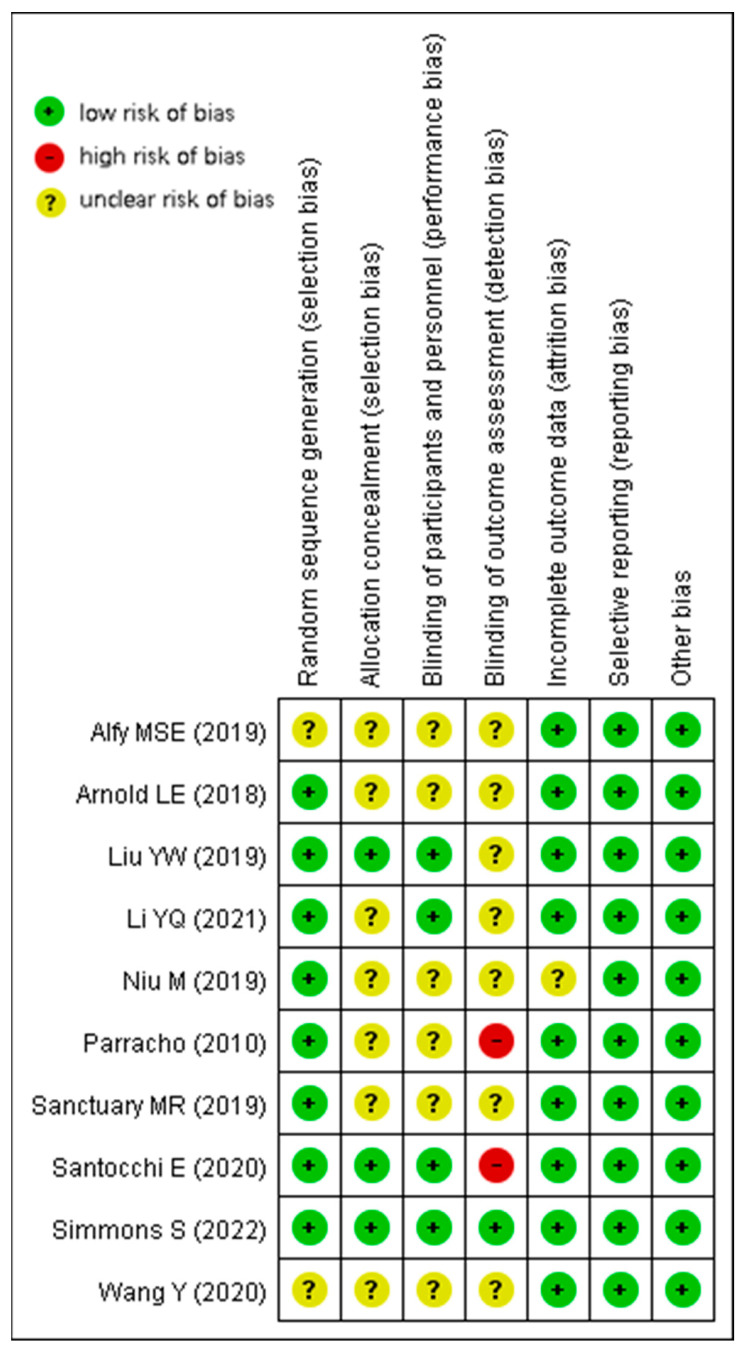
Risk of bias summary for RCTs and crossover controlled trials [23,24,25,26,27,28,29,30,31,32].

**Figure 4 nutrients-15-01415-f004:**
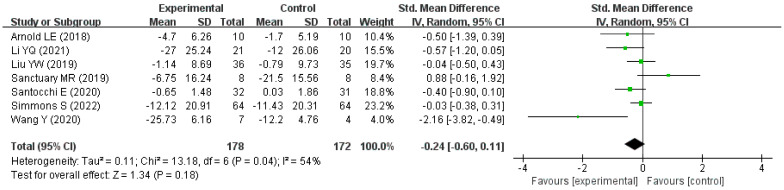
Forest plot for the effect of probiotics on improvement in autism-related behavioral symptoms [23,25,26,27,28,29,30].

**Figure 5 nutrients-15-01415-f005:**
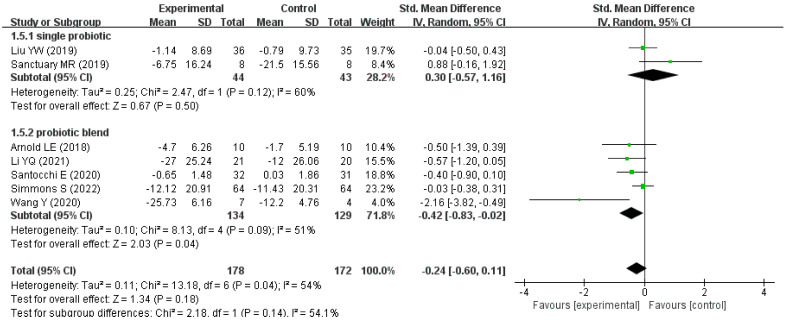
Forest plot for the subgroup analysis stratified by intervention measures [23,25,26,27,28,29,30].

**Figure 6 nutrients-15-01415-f006:**
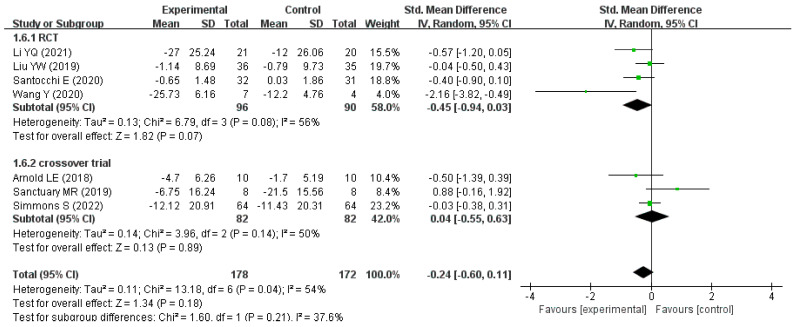
Forest plot for the subgroup analysis according to study type [23,25,26,27,28,29,30].

**Table 1 nutrients-15-01415-t001:** Basic characteristics of studies included in the systematic review.

Author (Year)	Country/Region	Type of Trial	Age (Year)	Sample Size	Intervention Group	Compare Group	Intervention Duration	Scale on Autism	GI Symptoms Are Measured
Niu M (2019) [24]	China	RCT	3–8	65	*n* = 37 probiotics +ABA 3.6 ×10^10^ CFU/day	*n* = 28 ABA	4 weeks	ATEC	Yes
Li YQ (2021) [25]	China	RCT	3–6	41	*n* = 21 *Bifidobacterium* triple live dispersion +ABA 9 × 10^7^ CFU/day	*n* = 20 ABA	3 months	ATEC	Yes
Liu YW (2019) [23]	Taiwan (China)	RCT	7–15	71	*n* = 36 *Lactobacillus plantarum* PS128 3 × 10^7^ CFU/day	*n* = 35 placebo	4 weeks	CGI-I ABC-T SRS SNAP-IV	No
Santocchi E (2020) [27]	Italy	RCT	1.5–6	63	*n* = 32 DSF 9 × 10^11^ CFU/day	*n* = 31 placebo	6 months	ADOS-CSS, GI Severity Index Score	Yes
Wang Y (2020) [26]	China	RCT	2–8	11	*n* = 7 probiotics * + FOS 10^10^ CFU/day	*n* = 4 placebo	108 days	ATEC 6-GSI	Yes
Arnold LE (2018) [28]	America	crossover controlled trials	3–12	10	*n* = 10 DSF 4.5–9 × 10^12^ CFU/day	*n* = 10 placebo	8 weeks	ABC, SRS	Yes
Parracho H (2010) [31]	Britain	crossover controlled trials	3–16	17	*n* = 17 *Lactobacillus plantarum* WCFS1 4.5 × 10^10^ CFU/day	*n* = 17 placebo	6 weeks	DBC	Yes
Sanctuary MR (2019) [29]	America	crossover controlled trials	2–11	8	*n* = 8 *Bifidobacterium infantis* +BCP 2 × 10^10^ CFU/day	*n* = 8 BCP	5 months	ABC, GIH	Yes
Simmons S (2022) [30]	America	crossover controlled trials	5–11	64	*n* = 64 DSF 4.5 × 10^11^ CFU/y	*n* = 64 placebo	12 weeks	ATEC GHI ABC	Yes
Alfy MSE (2019) [32]	Egypt	RCT	2–10	100	*n* = 50 Lacteol Fort 2 × 10^8^ CFU/day	*n* = 50 standard treatment	12 weeks	ATEC 6-GSI	Yes

ABA: applied behavior analysis; *Bifidobacterium* triple live dispersion: contains *Bifidobacterium longum*, *Lactobacillus acidophilus*, and *Enterococcus faecalis*; DSF: a patented mixture (containing 8 probiotic strains, each containing 450 billion lyophilized bacteria, including *Streptococcus thermophilus*, *Bifidobacterium brevis*, *Bifidobacterium longum*, *Bifidobacterium infantis*, *Lactobacillus acidophilus*, *Lactobacillus plantarum*, *Lactobacillus para-casei*, and *Lactobacillus delbrueckii* subsp. *Bulgaricus*) already approved for use in children, marketed as Vivomixx^®^ in EU, Visbiome^®^ in USA; FOS: fructo-oligosaccharide; BCP: bovine colostrum product; Lacteol Fort: a mixture of *Lactobacillus deuteri* and *Lactobacillus fermentum*; ATEC: autism treatment evaluation checklist; CGI-I: clinical global impression-improvement; ABC-T: autism behavior checklist-Taiwan version; SRS: social responsiveness scale; SNAP-IV: the Chinese version of the Swanson, Nolan, and Pelham-IV; ADOS-CSS: the total autism diagnostic observation schedule-calibrated severity score; 6-GSI: 6-GI severity index; ABC: aberrant behavior checklist; DBC: development behavior checklist; GIH: gastrointestinal history; *: a probiotic blends (*Bifidobacterium infantis* Bi-26, *Lactobacillus rhamnosus* HN001, *Bifidobacterium lactis* BL-04, and *Lactobacillus paracasei LPC-37*).

**Table 2 nutrients-15-01415-t002:** The characteristics of studies included in the meta-analysis.

Author (Year)	Country	Type of Trial	Sample Size (Intervention /Compare)	Age (Year)	Intervention Measure/Compare Measure	Intervention Duration	Change in Score (Intervention /Compare)	Scale
Li YQ (2021) [25]	China	RCT	21/20	3–6	*Bifidobacterium* triple live dispersion +ABA/ABA	3 months	−27.00 (25.24)/ −12.00 (26.06) *	ATEC
Liu YW (2019) [23]	Taiwan (China)	RCT	36/35	7–15	*Lactobacillus plantarum* PS128/Placebo	4 weeks	−1.14 (8.69)/ −0.79 (9.73) *	ABC-T
Santocchi E (2020) [27]	Italy	RCT	32/31	1.5–6	DSF/ Placebo	6 months	−0.65 (1.48)/ 0.03 (1.86) *	ADOS-CSS
Wang Y (2020) [26]	China	RCT	7/4	2–8	Probiotics + FOS/Placebo	108 days	−25.73 (6.16)/ −12.20 (4.76) *	ATEC
Arnold LE (2018) [28]	America	Crossover controlled trial	10/10	3–12	DSF/ Placebo	8 weeks	−4.70 (6.26)/ −1.70 (5.19)	SRS
Sanctuary MR (2019) [29]	America	Crossover controlled trial	8/8	2–11	*Bifidobacterium infantis* + BCP/BCP	5 weeks	−6.75 (16.24)/ −21.5 (15.56)	ABC
Simmons S (2022) [30]	America	Crossover controlled trial	64/64	5–11	DSF/ Placebo	12 weeks	−12.12 (20.91)/ −11.43 (20.31)	ATEC

ATEC: autism treatment evaluation checklist; ABC-T: autism behavior checklist-Taiwan version; ADOS-CSS: the total autism diagnostic observation schedule-calibrated severity score; SRS: social responsiveness scale; ABC: aberrant behavior checklist; *: standard deviation or mean values are not given in the original text and are calculated based on the data provided in Section 16.1.3.2 of the Cochrane handbook based on a correlation coefficient of 0.5.

## Data Availability

The datasets used in the present study can be available from the corresponding author upon reasonable request.

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
