# Peer review of "Effects of Probiotics on Autism Spectrum Disorder in Children: A Systematic Review and Meta-Analysis of Clinical Trials"

_nutrients, 2023, doi:10.3390/nu15061415_

Round 1

Reviewer 1 Report

Effect of probiotics on autism spectrum disorder in children: A systematic review and meta-analysis of clinical trials. 

After having read the manuscript, I have the following mayor comments:

-       Since their previous systematic review was published in 2022 and registered in PROSPERO in May 2021, why the current review does not include the study of Elisa et al (2020) that was included in the previous review? Why the previous review does not include Niu et al (2019), Alfi et al (2019), Santocchi et al (2020) since all of them are RCT?

-       In my opinion, the high number of articles screened (1058) compared to those finally included (12) indicates that the searching strategy could be improved. In addition, the reasons for exclusion of 1019 articles should be presented (for example grouping the articles by reason of exclusion). This information could be added as supplementary material.

-       Authors must include the specific strains and the CFU per dose. The effect of the probiotics is strain-specific, and therefore this information is essential. This should be an inclusion criteria (those articles that don’t give the strain information should be excluded).

-       Authors include 1 observational study (Mensi et al., 2021), which should be removed.

Other minor comments:

-       The original (updated) definition of probiotics (line 62) should be used (Nat Rev Gastroenterol Hepatol. 2017 Aug;14(8):491-502).

-       Microbial nomenclature should be written in italics. Please check the whole manuscript.

Author Response

On behalf of my co-authors, we thank you very much for giving us an opportunity to revise this paper. We appreciate you very much for their positive and constructive comments and suggestions on our manuscript entitled " Effect of probiotics on autism spectrum disorder in children: A systematic review and meta-analysis of clinical trials".
We carefully studied the comments of your and made revisions using the “Track Changes”. We have contacted the language editing company to make our language more standardized, but the language editing company has not finished modifying it for the time being, and we will send it to the editorial office as soon as the modification is completed. We have tried our best to revise our manuscript according to the comments. I hope this revised version is acceptable for publication. We would like to express our great appreciation to you for the comments on our paper.

Effect of probiotics on autism spectrum disorder in children: A systematic review and meta-analysis of clinical trials. 

After having read the manuscript, I have the following mayor comments:

1-       Since their previous systematic review was published in 2022 and registered in PROSPERO in May 2021, why the current review does not include the study of Elisa et al (2020) that was included in the previous review?

Response: In fact, this study has been included our review. Its correct citation from PubMed is as following:

Santocchi E, Guiducci L, Prosperi M, Calderoni S, Gaggini M, Apicella F, Tancredi R, Billeci L, Mastromarino P, Grossi E, Gastaldelli A, Morales MA, Muratori F. Effects of Probiotic Supplementation on Gastrointestinal, Sensory and Core Symptoms in Autism Spectrum Disorders: A Randomized Controlled Trial. Front Psychiatry. 2020 Sep 25;11:550593. doi: 10.3389/fpsyt.2020.550593. PMID: 33101079; PMCID: PMC7546872.

So, we describe its author as Santocchi et al (2020) in our review. But in the previous review of Song et al, its author was described as Elisa et al (2020), we think this is a mistake.

Song W, Zhang M, Teng L, Wang Y, Zhu L. Prebiotics and probiotics for autism spectrum disorder: a systematic review and meta-analysis of controlled clinical trials. J Med Microbiol. 2022 Apr;71(4). doi: 10.1099/jmm.0.001510. PMID: 35438624.

2.Why the previous review does not include Niu et al (2019), Alfi et al (2019), Santocchi et al (2020) since all of them are RCT?

Response: The study of Niu et al (2019) was not included in the previous review, because the data related to Meta analysis could not be extracted from this study. Actually, it was only included in our system review part, not in Meta analysis.

The study of Elisa et al (2020) (2020) in the previous review is in fact the study of Santocchi et al (2020) in our review, we have explained the reason in the question above.

The study of Alfi et al (2019) was retrieved from Medlink (The detailed search strategies are listed in Supplementary File S1.). The previous study probably did not search in this resource, because the previous study described their search strategies as following: “A systematic search of the literature was conducted in PubMed, Web of Science, Embase, and Cochrane Library to identify studies relevant to the current review.

3-       In my opinion, the high number of articles screened (1058) compared to those finally included (12) indicates that the searching strategy could be improved. In addition, the reasons for exclusion of 1019 articles should be presented (for example grouping the articles by reason of exclusion). This information could be added as supplementary material.

Response: Thanks for your kind suggestion. We improved our search strategy, and there were 913 articles screened after duplicated records removed, and the improved search strategy has been added to Supplementary File S1. And the reasons for exclusion of 876 articles were added to the PRISMA Flow Diagram (Figure 1. PRISMA Flow Diagram).

4.-       Authors must include the specific strains and the CFU per dose. The effect of the probiotics is strain-specific, and therefore this information is essential. This should be an inclusion criteria (those articles that don’t give the strain information should be excluded).

Response: Thanks for your kind suggestion. We have added the specific strains and the CFU per dose in Table 1.

5.-       Authors include 1 observational study (Mensi et al., 2021), which should be removed.

Response:Thanks for your kind suggestion. We reconsidered our inclusion criteria and now only included RCTs and cross-over trials. Two studies (Mensi et al., 2021and Shaaban et al., 2017) that did not meet the inclusion criteria were excluded.

Other minor comments:

6.-       The original (updated) definition of probiotics (line 62) should be used (Nat Rev Gastroenterol Hepatol. 2017 Aug;14(8):491-502).

Response: Thanks for your kind suggestion. But the literature you suggested presented the definition of prebiotics, not probiotics. We searched another expert consensus document on the definition of probiotics, and revised our manuscript based on it.

Line62-63: Probiotics are live microorganisms that, when administered in adequate amounts, confer a health benefit on the host.

Hill C, Guarner F, Reid G, Gibson GR, Merenstein DJ, Pot B, Morelli L, Canani RB, Flint HJ, Salminen S, Calder PC, Sanders ME. Expert consensus document. The International Scientific Association for Probiotics and Prebiotics consensus statement on the scope and appropriate use of the term probiotic. Nat Rev Gastroenterol Hepatol. 2014 Aug;11(8):506-14. doi: 10.1038/nrgastro.2014.66. Epub 2014 Jun 10. PMID: 24912386.

7.-       Microbial nomenclature should be written in italics. Please check the whole manuscript.

Response: Thanks for your kind suggestion. We have revised and check the whole manuscript.

Reviewer 2 Report

The authors describe a thorough systematic review and meta analysis of the efficacy of probiotic therapy on symptoms of ASD. The research design follows standard, accepted methods and the results are appropriately presented in plots and tables.

Suggestions:

The manuscript requires editing for proper English. Articles (the, a) are missing is some sentences. Also check singular/plural use.

Introduction, line 62: Describe some of the main microbial metabolites thought to be involved in the gut-brain axis and neurological function.

Results: Avoid the use of contractions.

Discussion: Perhaps just before "strengths and limitations" add a description of the current mechanisms proposed to explain how probiotics and their metabolites could improve ASD symptoms.

Author Response

On behalf of my co-authors, we thank you very much for giving us an opportunity to revise this paper. We appreciate you very much for their positive and constructive comments and suggestions on our manuscript entitled " Effect of probiotics on autism spectrum disorder in children: A systematic review and meta-analysis of clinical trials".
We carefully studied the comments of your and made revisions using the “Track Changes”. We have contacted the language editing company to make our language more standardized, but the language editing company has not finished modifying it for the time being, and we will send it to the editorial office as soon as the modification is completed. We have tried our best to revise our manuscript according to the comments. I hope this revised version is acceptable for publication. We would like to express our great appreciation to you for the comments on our paper.

The authors describe a thorough systematic review and meta analysis of the efficacy of probiotic therapy on symptoms of ASD. The research design follows standard, accepted methods and the results are appropriately presented in plots and tables.

Suggestions:

  1. The manuscript requires editing for proper English. Articles (the, a) are missing is some sentences. Also check singular/plural use.

Response: Thanks for your kind suggestion. We had our manuscript edited by AJE to improve the English expression . We will send back the edited manuscript to you as soon as we get it.

  1. Introduction, line 62: Describe some of the main microbial metabolites thought to be involved in the gut-brain axis and neurological function.

Response: Thanks for your kind suggestion. We have done.

Line63-66: Because they produce and transport neuroactive substances and act on the gut-brain axis, such as, γ-aminobutyric acid (GABA) produced by Lactobacillus brevis and Bifidobacterium denticola, and dopamine produced by Staphylococcus aureus and Escherichia coli , Dinan et al. defined them as Psychobiotics.

  1. Results: Avoid the use of contractions.

Response: Thanks for your kind suggestion. In results part, most contractions or abbreviations are related to the titles of scales on autism or the preparations of probiotics, which are repeated in the manuscript. We reduced the use of contractions in several places, but in order to be concise and succinct in writing, we still used most of these abbreviations. Please understand us, we have not accepted your suggestion completely.

  1. Discussion: Perhaps just before "strengths and limitations" add a description of the current mechanisms proposed to explain how probiotics and their metabolites could improve ASD symptoms.

Response: Thanks for your kind suggestion. We have explained how probiotics and their metabolites could improve ASD symptoms in discussion part, Line355-358.

Round 2

Reviewer 1 Report

The manuscript has improved. However I still have one important concern. Authors include the name of the probiotic strains only in 2 of the articles selected (Liu YW and Parracho). For the remaining articles, authors include the species name (Li, Santocchi, Arnold, Sanctuary, Simmons, Alfi) or just "probiotics". This information is of a great importance and should be given. In the case that this information is not given in the articles, this should be presented as a limitation.

Furthermore, the fact that authors excluded 165 articles because they were not carried out in humans, indicates that the searching strings could be improved by using additional filters.

Author Response

On behalf of my co-authors, we thank you very much for giving us an opportunity to revise this paper. We appreciate you very much for their positive and constructive comments and suggestions on our manuscript entitled " Effect of probiotics on autism spectrum disorder in children: A systematic review and meta-analysis of clinical trials".
We carefully studied the comments of your and made revisions using the “Track Changes”. We have contacted the language editing company to make our language more standardized. We have tried our best to revise our manuscript according to the comments. I hope this revised version is acceptable for publication. We would like to express our great appreciation to you for the comments on our paper.

The manuscript has improved. However I still have one important concern.

  1. Authors include the name of the probiotic strains only in 2 of the articles selected (Liu YW and Parracho). For the remaining articles, authors include the species name (Li, Santocchi, Arnold, Sanctuary, Simmons, Alfi) or just "probiotics". This information is of a great importance and should be given. In the case that this information is not given in the articles, this should be presented as a limitation.

Response: Thanks for your kind suggestion. In fact, only one article (Niu M et al.) did not give the name of the probiotic strains used, so we did not include this article in the meta-analysis, and the names of the probiotic strains used in the remaining articles have been added in Table 2.  Probiotic blends were used in the studies of Li YQ(2021), Liu YW(2019),Santocchi E(2020),Wang Y(2020), Arnold LE (2018)  Simmons S (2022) , and Alfy MSE (2019), so all the strains in each blends were listed in the notation under the Talbe 2.

We have pointed the limitation that the strains used were not reported in the study of Niu M et al., in Line184-185.

We added the exclusion criteria of meta-analysis: “the information on probiotic strains was not available.” in Line 105-106.

2.Furthermore, the fact that authors excluded 165 articles because they were not carried out in humans, indicates that the searching strings could be improved by using additional filters.

Response: Thanks for your kind suggestion. We improved our search strategy, and there were 676 articles screened after duplicated records removed, and the improved search strategy has been added to Supplementary File S1. And the Articles that were not carried out in humans have been reduced to 44.
